# Synthesis and Evaluation on the Fungicidal Activity of S-Alkyl Substituted Thioglycolurils

**DOI:** 10.3390/ijms24065756

**Published:** 2023-03-17

**Authors:** Ekaterina E. Vinogradova, Anna L. Alekseenko, Sergey V. Popkov, Natalya G. Kolotyrkina, Angelina N. Kravchenko, Galina A. Gazieva

**Affiliations:** 1N. D. Zelinsky Institute of Organic Chemistry, Russian Academy of Sciences, 47 Leninsky Prosp., Moscow 119991, Russia; 2Faculty of Chemical-Pharmaceutical Technologies and Biomedical Preparations, Mendeleev University of Chemical Technology, 9 Miusskaya Sq., Moscow 125047, Russia

**Keywords:** tetrahydroimidazo[4,5-*d*]imidazol-2(1*H*)-ones, thioglycolurils, S-alkyl thioglycolurils, fungicidal activity, phytopathogenic fungi, *Candida albicans*

## Abstract

A series of S-alkyl substituted thioglycolurils was prepared through the alkylation of corresponding thioglycolurils with halogenoalkanes and tested for their fungicidal activity against six phytopathogenic fungi from different taxonomic classes: *Venturia inaequalis*, *Rhizoctonia solani*, *Fusarium oxysporum*, *Fusarium moniliforme*, *Bipolaris sorokiniana*, and *Sclerotinia sclerotiorum,* and two pathogenic yeasts: *Candida albicans* and *Cryptococcus neoformans var. grubii*. A number of S-alkyl substituted thioglycolurils exhibited high activity against *Venturia inaequalis* and *Rhizoctonia solani* (85–100% mycelium growth inhibition), and moderate activity against other phytopathogens. S-Ethyl substituted thioglycolurils possessed a high activity against *Candida albicans*. Additionally, the hemolytic and cytotoxic properties of promising derivatives were determined using human red blood cells and human embryonic kidney cells, respectively. Two S-ethyl derivatives possessed both low cytotoxicity against normal human cells and high fungicidal activity against *Candida albicans*.

## 1. Introduction

Parasitic fungi endanger the health of humans or domestic animals and cause damage to crops or ornamental plants. Invasive fungal infections are increasingly recognized as life-threatening infections in clinics. More than 300 million people suffer from fungal infections, which lead to over 1,350,000 deaths per year [1]. The most common pathogens of mycoses in humans are *Aspergillus* spp., *Candida* spp., and *Cryptococcus* spp. [1,2]. In addition, plant diseases are an important factor in agricultural production, causing significant economic losses [3,4]. Antifungal resistance represents a major challenge and requires a constant updating of the drugs and agrochemicals used [5,6]. In this regard, the development of new effective fungicides is an urgent task [7]. 

Imidazole and benzimidazole cores are widely found in natural products and biologically active compounds [8,9,10,11,12,13,14]. 2-Thioderivatives of imidazole were found to possess antifungal activity [15,16,17,18,19,20,21]. 4-Methyl-2-methylthio-4-phenyl-1-phenylamino-1*H*-imidazol-5(4*H*)-one, known as the fungicide *fenamidone*, has been present on the market of agrochemical plant protection products since 2001. It blocks fungal energy production due to mitochondrial respiration inhibition through binding to the Qo site of the cytochrome bc1 complex [2,17]. Its analogues (for example **1**) were found to show fungicidal activity against *Phytophthora infestans*, *Botrytis cinerea*, *Pyrcularia oryzae,* and *Fusarium oxysporum* (Figure 1) [18,19]. 

Recently [20], we have synthesized a series of S-methyl and S-ethyl thioglycoluril derivatives **2**, **3** possessing fungicidal activity against the phytopatogenes *Rhizoctonia solani, Fusarium oxysporum, Fusarium moniliforme*, and *Bipolaris sorokiniana* (see Figure 1). It was found that S-ethyl derivatives are somewhat more active than S-methyl ones. We assumed that the elongation of the alkyl chain at the S atom from C2 to C3 and C4 would lead to an increase in fungicidal activity. In this paper, we synthesized a series of new S-propyl, S-allyl and S-butyl derivatives of 4-[(*E*)-((*E*)-3-phenylallylidene)amino]-5-thioxohexahydroimidazo[4,5-*d*]imidazol-2(1*H*)-one (4-[(*E*)-((*E*)-3-phenylallylidene)amino]thioglycoluril) and evaluated their fungicidal activity. S-allyl thioglycolurils were synthesized to reveal the influence of the unsaturated fragment on the activity.

## 2. Results and Discussion

### 2.1. Chemistry

The synthetic route to the S-alkyl thioglycolurils **4a–f**, **5a–f**, and **6a–h** is outlined in Figure 1. To begin, imidazotriazines **7a,b** were synthesized via the cyclisation of 4,5-dihydroxyimidazolidin-2-ones **8a,b** with thiosemicarbazide [22]; the treatment of compounds **7a,b** with (*E*)-3-phenyl(furan-2-yl)acrylaldehyde derivatives **9a–d** in refluxing methanol in the presence of catalytic HCl gave thioglycolurils **10a–h** in 43–65% yields [20]. Alkylthio derivatives **4–6** were prepared through the alkylation of corresponding thioglycolurils **10** with 1-bromopropane, 3-bromoprop-1-ene, or 1-bromobutane, respectively, in the presence of potassium carbonate (Figure 1).

The structures of compounds **4a–f**, **5a–f**, and **4a–f** were confirmed via IR, ^1^H NMR, ^13^C NMR, and HRMS spectral data. The starting compounds **10a–h** have an *E* configuration around the C=N and C=C bonds [20]. The constants of the vicinal interaction of protons of the CH=CH-Ar fragment of derivatives **4–6** lie in the range of 15.8–16.1 Hz, which is also characteristic of the *trans* arrangement of substituents at a double bond.

### 2.2. Fungicidal Activity Testing

Synthesized alkylthio derivatives **4–6** were tested in vitro to a common conventional procedure [20,23,24,25,26] with six phytopathogenic fungi characterized by high impact on crop production: *Venturia inaequalis*, *Rhizoctonia solani*, *Fusarium oxysporum*, *Fusarium moniliforme*, *Bipolaris sorokiniana*, and *Sclerotinia sclerotiorum*. The effect of the testing compounds on the mycelium radial growth in potato-saccharose agar was measured at a concentration 30 μg mL^−1^. Triadimefon was used as a reference compound (Table 1).

As shown in Table 1, some compounds demonstrated moderate to excellent mycelial growth inhibition. In particular, compounds **4a,b,e, 5e,f,** and **6a,b,e,f,g** showed moderate inhibitory activity against the causative agent of apple scab disease, *V. inaequalis*, with inhibition percentages of 42–75%, which were comparable or higher than those for triadimefon (41%). Derivative **6h** possessed excellent fungicidal activity, suppressing the growth of *V. inaequalis* mycelium by 100%. Compounds **4b,e, 5e,** and **6b,c,d,e,f,h** inhibited the growth of *R. solani* mycelium by 43–78%, which was comparable or higher than that of triadimefon (43%). At the same time, derivatives **4a** (100%), **5f** (85%), **6a** (91%), and **6g** (93%) almost totally inhibited the growth of the fungus. Compounds **4a,e, 5f,** and **6a,b,c,e,g** exhibited a remarkable antifungal activity against *F. oxysporum* with inhibition rates of 51–75%, although these values were lower than that of triadimefon (77%). Compounds **4a**,**b** and **6a**,**b**,**e–h** showed remarkable activity against *F. moniliforme* (55–66% mycelium growth inhibition). However, they are inferior to the triadimefon (87%). The inhibitory effect of compounds **4a**,**b**,**d**,**e**, **5e**,**f**, and **6a**,**b**,**d**,**e–h** (from 46 to 75%) against *B. sorokiniana* exceeded that of triadimefon (44%). Compounds **4a**,**e**, **5f**, and **6a**,**e**,**g** showed activity at the same or slightly less (from 53 to 62%) than the triadimefon level (61%) against *S. sclerotiorum*. 

From the SAR point of view, compounds **4–6** differ in substituents at N(1) and N(3) nitrogen atoms and at the sulfur atom, as well as an aromatic fragment. Substituents at N(1) and N(3) nitrogen atoms do not have a definite effect on the activity of compounds. In some cases, 1,3-dimethyl substituted derivatives **4a, 6a,** and **6e** were more active than the corresponding 1,3-diethyl derivatives **4b, 6b,** and **6f**. At the same time, 1,3-diethyl substituted compounds **5f** and **6h** were more active than the corresponding 1,3-dimethyl derivatives **5e** and **6g**. In all other pairs of compounds, it is impossible to definitely choose which compound is better.

Among the S-alkyl derivatives, in general, activity increased with an increase in the length of the alkyl chain (Figure 2, Table 1), which is quite likely due to the increasing lipophilicity with the elongation of the alkyl chain. The activity of S-allyl derivatives **5c–e** was at or slightly less than the level of S-propyl derivatives **4c–e** (Table 1).

The aromatic fragment also affected the activity of the tested compounds. Compounds containing a phenyl ring with an electron-donating methoxy group possessed moderate to remarkable activity (**4e, 5e, 5f, 6e, 6f**). Introducing an electron-withdrawing nitro group in the ortho-position of the aromatic fragment led to a significant reduction in activity (**4c,d, 5c,d, 6c,d**). The most active compounds are among the derivatives containing an unsubstituted phenyl or furyl ring (**4a, 6a, 6g, 6h**) (Figure 3, Table 1). Perhaps, besides the electron effects of the aryl ring substituents, the steric ones can affect the activity.

Thus, we found some structure–activity correlations: (i) an increase in the length of the alkyl chain at the sulfur atom leads to an increase in activity; and (ii) depending on the arylmethylidene fragments, compounds could be arranged according to decreasing activity as follows: Ph-substituted ≥ Fu-substituted > 2-MeOC_6_H_4_-substituted >> 2-NO_2_C_6_H_4_- substituted.

Some S-alkyl derivatives, both new and earlier synthesized [20], were assessed for microbiological activity by COADD (the Community for Antimicrobial Drug Discovery) [27,28,29,30,31]. S-Methyl (**2a,b,e**), S-ethyl (**3a,b,d,e,g**), S-propyl (**4a,d**) and S-butyl thioglycolurils (**6b**) were tested for inhibitory activity against two yeast fungi (*Candida albicans ATCC 90028* and *Cryptococcus neoformans var. grubii ATCC 208821*). All the tested compounds did not possess significant activity against *Cryptococcus neoformans* in a concentration of 32 μg mL^−1^ (<50% mycelium growth inhibition, MIC values >32 μg mL^−1^ for **3a,b,e,g**). The results of the analysis of activity against *Candida albicans* are shown in Table 2.

Among the tested S-alkyl derivatives, only S-ethyl thioglycolurils **3a,b,e,g** exhibited high fungicidal activity against *Candida albicans*. S-Methyl **2a,b,e**, S-propyl **4a,d,** S-butyl thioglycolurils **6b**, and S-ethyl thioglycoluril **3d** with a nitrophenyl ring as an aromatic fragment were inactive. For the more potent compounds, **3a,b,e,g**, minimum inhibitory concentrations and cytotoxicity on human embryonic kidney cells (HEK-293, ATCC CRL-1573, CC_50_) and human red blood cells (RBC, HC_10_) were additionally determined. Compounds **3b** and **3e** were found to exhibit higher cytotoxicity than fungicidal activity, while two compounds, **3a** and **3g,** displayed potent activity along with low cytotoxicity towards HEK-293 and RBC cells.

### 2.3. Cytotoxicity Assay

The cytotoxic activity of some new compounds, **4b** and **5d–f**, was studied at the concentration 10^−5^ M against a panel of approximately 60 cancer cell lines derived from nine neoplastic diseases (leukemia, melanoma, lung, colon, CNS, ovarian, renal, prostate, and breast cancers) using the sulforhodamine B method of the National Cancer Institute Developmental Therapeutic Program (DTP). Some results are presented in Table 3. It was found that compounds **4b** and **5d,e** did not exhibit cytotoxic activity against all tested cancer cell lines. The mean growth of cell lines was 94.44–95.93%. Compound **5f** in a 10 μM concentration was slightly more active inhibiting the cell growth of HL-60(TB), K562, SR (leukemia), MDA-MB-435 (melanoma), and MCF-7, MDA-MB-468 (breast cancer). The growth percentage values for these cell lines were 2.79–45.58 %. On the whole, it can be concluded that synthesized compounds **4b** and **5d–f** practically do not possess cytotoxicity.

## 3. Materials and Methods

### 3.1. Chemistry

#### 3.1.1. General Information

Melting points were determined in open glass capillaries on a Gallenkamp (Sanyo) melting point apparatus. IR spectra were recorded on a Bruker ALPHA instrument on KBr pellets. High resolution mass spectra (HRMS) were measured on a Bruker micrOTOF II instrument using electrospray ionization (ESI). The measurements were completed in a positive ion mode (interface capillary voltage—4500 V) or in a negative ion mode (3200 V), with a mass range from *m*/*z* 50 to *m*/*z* 3000 Da, and external or internal calibration was conducted with electrospray calibrant solution (Fluka). A syringe injection was used for solutions in acetonitrile or methanol (flow rate 3 μL/min). Nitrogen was applied as a dry gas; the interface temperature was set at 180 °C. ^1^H and ^13^C NMR spectra were recorded on Bruker AV-300 (300.13 MHz (1H)) and Bruker AM-300 (300.13 and 75.47 MHz, respectively) spectrometers using DMSO-*d*_6_ as a solvent and referenced to the residual solvent peak. The chemical shifts are reported in ppm (δ); multiplicities are indicated by s (singlet), d (doublet), t (triplet), m (multiplet). Coupling constants, J, are reported in hertz. The analysis of compound **4f** was carried out on a 1100 LS/MSD (Agilent Technologies) chromate-mass spectrometer equipped with an ELSD PL-ELS-1000 mass detector, with detection at 254 nm. The column used was Onyx monolithic C18, 50 × 4.6 mm. The flow rate was 3.75 mL min^–1^ and the eluent gradient was “A” (2.0 min)—“B” (0.6 min)—“A” (0.2 min) (A—0.1% F_3_CCOOH, 2.5% MeCN in H_2_O, B—0.1% F_3_CCOOH in MeCN).

1-Bromopropane, 99%; 3-bromoprop-1-ene, 99%, stabilized; and 1-bromobutane, 99%, were ordered from Acros organics. Thioglycolurils **10a–h**, and S-methyl and S-ethyl derivatives **2a,b,e,** and **3a,b,d,e,g** were prepared according to the procedures in the literature [20].

#### 3.1.2. General Procedure for the Synthesis of Alkylthio Derivatives of Thioglycoluril **4a–f**, **5a–f**, and **4a–f**

To a stirred suspension of thioglycoluril (**10**) (1 mmol) and potassium carbonate (0.138 g, 1 mmol) in methanol (30 mL), the corresponding bromoalkane (2.5 mmol) was added. The resulting mixture was stirred at 60 °C for 24 or 4 h (for the synthesis of compounds **4** or **5**, **6**, respectively), and then concentrated to dryness. A precipitate was washed with water on the filter and dried. Recrystallization from MeOH: H_2_O (1: 1) gave the S-alkyl derivative (**4a–f**, **5a–f**, **6a–h**).

**1,3-Dimethyl-4-[((1*E*,2*E*)-3-phenylallylidene)amino]-5-propylthio-3,3a,4,6a-tetrahydroimidazo[4,5-*d*]imidazol-2(1*H*)-one (4a).** Yield 189 mg (53%); white solid; mp: 155–157 °C. IR (KBr): ν 3056, 3030 (Ar), 2997, 2981, 2961, 2930 (Alk), 2873, 2863 (C-S), 1698, 1562 (C=O, C=N) cm^–1^. ^1^H NMR (300 MHz, DMSO-*d*_6_): δ 0.96 (t, *J* = 7.3 Hz, 3H, Me), 1.55–1.75 (m, 2H, CH_2_), 2.81 (s, 3H, NMe), 2.89 (s, 3H, NMe), 2.99 (t, *J* = 7.1 Hz, 2H, SCH_2_), 5.55 (d, *J* = 7.7 Hz, 1H, CH), 5.87 (d, *J* = 7.8 Hz, 1H, CH), 6.96 (dd, *J* = 15.9, 8.5 Hz, 1H, =CH), 7.05 (d, *J* = 16.0 Hz, 1H, Ph-CH=), 7.25–7.45 (m, 3H, Ph-3-5), 7.58 (d, *J* = 7.4 Hz, 2H, Ph-2,6), 7.93 (d, *J* = 8.5 Hz, 1H, N=CH). ^13^C NMR (75 MHz, DMSO-*d*_6_): δ 13.31 (Me), 22.23 (CH_2_), 28.46 (SCH_2_), 30.79 (NMe), 32.00 (NMe), 72.06 (CH), 79.94 (CH), 125.45, 127.03, 128.79, 129.02, 136.07, 137.86 (=CH, Ph), 141.26 (HC=N), 157.89 (C=O), 165.91 (C=N). HRMS (ESI): Calculated for C_18_H_23_N_5_OS [M + H]^+^: 358.1696; found: 358.1695.

**1,3-Diethyl-4[((1*E*,2*E*)-3-phenylallylidene)amino]-5-propylthio-3,3a,4,6a-tetrahydroimidazo[4,5-*d*]imidazol-2(1*H*)-one (4b).** Yield 235 mg (61%); white solid; mp: 132–134 °C. IR (KBr): ν 3083, 3033 (Ar), 2964, 2930 (Alk), 2871 (C-S), 1979, 1879, 1821 (Ar), 1706, 1565 (C=O, C=N) cm^–1^. ^1^H NMR (300 MHz, DMSO-*d*_6_): δ 0.90–1.05 (m, 6H, 2Me), 1.13 (t, *J* = 7.1 Hz, 3H, Me), 1.50–1.80 (m, 2H, CH_2_), 2.85–3.10 (m, 2H, SCH_2_), 3.12–3.29 (m, 3H, NCH_2_), 3.35–3.55 (m, 1H, NCH_2_), 5.65 (d, *J* = 7.8 Hz, 1H, CH), 5.93 (d, *J* = 7.9 Hz, 1H, CH), 6.90–7.10 (m, 2H, =CH, Ph-CH=), 7.20–7.45 (m, 3H, Ph-3-5), 7.60 (d, *J* = 7.4 Hz, 2H, Ph-2,6), 7.78 (d, *J* = 7.8 Hz, 1H, N=CH). ^13^C NMR (75 MHz, DMSO-*d*_6_): δ 13.17 (Me), 13.44 (Me), 13.87 (Me), 22.17 (CH_2_), 31.85 (SCH_2_), 36.25 (NCH_2_), 37.64 (NCH_2_), 70.37 (CH), 78.59 (CH), 125.33, 126.91, 128.59, 128.81, 135.93, 137.47 (=CH, Ph), 140.48 (HC=N), 157.02 (C=O), 165.21 (C=N). HRMS (ESI): Calculated for C_20_H_27_N_5_OS [M + H]^+^: 386.2009; found: 386.2006.

**1,3-Dimethyl-4-{[(1*E*,2*E*)-3-(2-nitrophenyl)allylidene]amino}-5-propylthio-3,3a,4,6a-tetrahydroimidazo[4,5-*d*]imidazol-2(1*H*)-one (4c).** Yield 330 mg (82%); yellow solid; mp: 147–149 °C. IR (KBr): ν 2968, 2935 (Alk), 2871 (C-S), 1699, 1564 (C=O, C=N) cm^–1^. ^1^H NMR (300 MHz, DMSO-*d*_6_): δ 0.96 (t, *J* = 7.3 Hz, 3H, Me), 1.50–1.80 (m, 2H, CH_2_), 2.81 (s, 3H, NMe), 2.88 (s, 3H, NMe), 3.00 (t, *J* = 7.1 Hz, 2H, SCH_2_), 5.57 (d, *J* = 7.6 Hz, 1H, CH), 5.90 (d, *J* = 7.8 Hz, 1H, CH), 7.04 (dd, *J* = 15.7, 8.8 Hz, 1H, =CH), 7.32 (d, *J* = 15.7 Hz, 1H, Ph-CH=), 7.55 (t, *J* = 7.7 Hz, 1H, Ar), 7.70 (t, *J* = 7.7 Hz, 1H, Ar), 7.96 (d, *J* = 8.1 Hz, 1H, N=CH), 8.00–8.10 (m, 2H, Ar). ^13^C NMR (75 MHz, DMSO-*d*_6_): δ 13.12 (Me), 22.04 (CH_2_), 28.24 (CH_2_), 30.48 (NMe), 31.86 (NMe), 71.67 (CH), 79.75 (CH), 124.32, 127.90, 129.20, 130.16, 130.33, 130.43, 133.26 (=CH, Ar), 140.05 (HC=N), 147.89 (ArC-NO_2_), 157.55 (C=O), 165.39 (C=N). HRMS (ESI): Calculated for C_18_H_22_N_6_O_3_S [M + H]^+^: 403.1547; found: 403.1541.

**1,3-Diethyl-4-{[(1*E*,2*E*)-3-(2-nitrophenyl)allylidene]amino}-5-propylthio-3,3a,4,6a-tetrahydroimidazo[4,5-*d*]imidazol-2(1*H*)-one (4d).** Yield 301 mg (70%); yellow solid; mp: 129–131 °C. IR (KBr): ν 2972, 2932, 2918 (Alk), 2871 (C-S), 1691, 1563 (C=O, C=N) cm^–1^. ^1^H NMR (300 MHz, DMSO-*d*_6_): δ 0.85–1.05 (m, 6H, 2Me), 1.13 (t, *J* = 7.1 Hz, 3H, Me), 1.50–1.80 (m, 2H, CH_2_), 2.90–3.10 (m, 2H, SCH_2_), 3.12–3.30 (m, 3H, NCH_2_), 3.40–3.55 (m, 1H, NCH_2_), 5.66 (d, *J* = 7.7 Hz, 1H, CH), 5.95 (d, *J* = 7.9 Hz, 1H, CH), 7.06 (dd, *J* = 15.8, 8.8 Hz, 1H, =CH), 7.30 (d, *J* = 15.7 Hz, 1H, Ph-CH=), 7.55 (t, *J* = 7.7 Hz, 1H, Ar), 7.71 (t, *J* = 7.7 Hz, 1H, Ar), 7.87 (d, *J* = 8.9 Hz, 1H, N=CH), 7.97 (d, *J* = 8.2 Hz, 1H, Ar), 8.03 (d, *J* = 7.9 Hz, 1H, Ar). ^13^C NMR (75 MHz, DMSO-*d*_6_): δ 13.15 (Me), 13.43 (Me), 13.94 (Me), 22.13 (CH_2_), 31.86 (SCH_2_), 36.23 (NCH_2_), 37.74 (NCH_2_), 70.34 (CH), 78.58 (CH), 124.40, 127.99, 129.25, 130.16, 130.23, 130.58, 133.33 (=CH, Ar), 139.62 (HC=N), 147.85 (ArC-NO_2_), 157.02 (C=O), 164.92 (C=N). HRMS (ESI): Calculated for C_20_H_26_N_6_O_3_S [M + H]^+^: 431.1860; found: 431.1855.

**4-{[(1*E*,2*E*)-3-(2-Methoxyphenyl)allylidene]amino}-1,3-dimethyl-5-propylthio-3,3a,4,6a-tetrahydroimidazo[4,5-*d*]imidazol-2(1*H*)-one (4e).** Yield 197 mg (51%); white solid; mp: 154–156 °C. IR (KBr): ν 2997, 2964, 2924 (Alk), 2874, 2838 (C-S), 1705, 1560 (C=O, C=N) cm^–1^. ^1^H NMR (300 MHz, DMSO-*d*_6_): δ 0.95 (t, *J* = 7.3 Hz, 3H, Me), 1.50–1.80 (m, 2H, CH_2_), 2.80 (s, 3H, NMe), 2.89 (s, 3H, NMe), 2.99 (t, *J* = 7.1 Hz, 2H, SCH_2_), 3.85 (s, 3H, OMe), 5.55 (d, *J* = 7.7 Hz, 1H, CH), 5.88 (d, *J* = 7.8 Hz, 1H, CH), 6.80–6.99 (m, 2H, Ar, =CH), 7.05 (d, *J* = 8.3 Hz, 1H, Ar), 7.20–7.40 (m, 2H, Ar, Ph-CH=), 7.64 (d, *J* = 7.5 Hz, 1H, Ar), 7.98 (d, *J* = 9.0 Hz, 1H, N=CH). ^13^C NMR (75 MHz, DMSO-*d*_6_): δ 13.16 (Me), 22.07 (CH_2_), 28.26 (NMe), 30.49 (NMe), 31.82 (SCH_2_), 55.47 (OMe), 71.76 (CH), 79.70 (CH), 111.50, 120.71, 124.30, 125.75, 126.92, 129.94, 132.20 (=CH, Ar), 141.83 (HC=N), 156.63 (ArC-OMe), 157.61 (C=O), 165.60 (C=N). HRMS (ESI): Calculated for C_19_H_25_N_5_O_2_S [M + H]^+^: 388.1802; found: 388.1811.

**1,3-Diethyl-4-{[(1*E*,2*E*)-3-(2-methoxyphenyl)allylidene]amino}-5-propylthio-3,3a,4,6a-tetrahydroimidazo[4,5-*d*]imidazol-2(1*H*)-one (4f).** Yield mg (90%); white solid; mp: 145–147 °C. IR (KBr): ν 3083, 3033 (Ar), 2964, 2930 (Alk), 2871 (C-S), 1979, 1879, 1821 (Ar), 1706, 1565 (C=O, C=N) cm^–1^. ^1^H NMR (300 MHz, DMSO-*d*_6_): δ 0.80–1.05 (m, 6H, 2Me), 1.13 (t, *J* = 7.0 Hz, 3H, Me), 1.50–1.80 (m, 2H, CH_2_), 2.85–3.10 (m, 2H, SCH_2_), 3.15–3.30 (m, 3H, NCH_2_), 3.40–3.50 (m, 1H, NCH_2_), 3.85 (s, 3H, OMe), 5.65 (d, *J* = 7.8 Hz, 1H, CH), 5.93 (d, *J* = 7.9 Hz, 1H, CH), 6.88–7.00 (m, 2H, Ar, =CH), 7.05 (d, *J* = 8.4 Hz, 1H, Ar), 7.15–7.40 (m, 2H, Ar, Ph-CH=), 7.64 (d, *J* = 7.3 Hz, 1H, Ar), 7.83 (d, *J* = 8.9 Hz, 1H, N=CH). ^13^C NMR (75 MHz, DMSO-*d*_6_): δ 13.19 (Me), 13.44 (Me), 13.90 (Me), 22.20 (CH_2_), 31.88 (SCH_2_), 36.27 (NCH_2_), 37.67 (NCH_2_), 55.53 (OMe), 70.37 (CH), 78.58 (CH), 111.54, 120.76, 124.31, 125.77, 127.10, 130.01, 132.20 (=CH, Ar), 141.37 (HC=N), 156.70 (ArC-OMe), 157.11 (C=O), 165.26 (C=N). LS MS: retention time 1.481 min (95%), API-ES, m/z 415.9 [M]^+^.

**5-Allylthio-1,3-dimethyl-4-[((1*E*,2*E*)-3-phenylallylidene)amino]-3,3a,4,6a-tetrahydroimidazo[4,5-*d*]imidazol-2(1*H*)-one (5a).** Yield 209 mg (59%); white solid; mp: 149–151 °C. IR (KBr): ν 3078, 3045 (Ar, =CH_2_), 2992, 2939 (Alk), 2879 (C-S), 1947, 1872 (Ar), 1718, 1560 (C=O, C=N) cm^–1^. ^1^H NMR (300 MHz, DMSO-*d*_6_): δ 2.81 (s, 3H, NMe), 2.89 (s, 3H, NMe), 3.40–3.80 (m, 2H, SCH_2_), 5.13 (d, *J* = 10.0 Hz, 1H, =CH_2_), 5.29 (d, *J* = 17.1 Hz, 1H, =CH_2_), 5.57 (d, *J* = 7.8 Hz, 1H, CH), 5.80–6.10 (m, 2H, CH, -CH=), 6.93 (dd, *J* = 16.0, 8.6 Hz, 1H, =CH), 7.06 (d, *J* = 16.0 Hz, Ph-CH=), 7.20–7.45 (m, 3H, Ph-3-5), 7.59 (d, *J* = 7.5 Hz, 2H, Ph-2,6), 7.94 (d, *J* = 8.6 Hz, 1H, N=CH). ^13^C NMR (75 MHz, DMSO-*d*_6_): δ 28.25 (NMe), 30.58 (NMe), 32.56 (SCH_2_), 72.06 (CH), 79.82 (CH), 118.22 (=CH_2_), 125.33, 126.89, 128.59, 128.81, 133.43, 135.92, 137.72 (=CH, Ph), 141.19 (HC=N), 157.60 (C=O), 165.12 (C=N). HRMS (ESI): Calculated for C_18_H_21_N_5_OS [M + H]^+^: 356.1540; found: 356.1546.

**5-Allylthio-1,3-diethyl-4-[((1*E*,2*E*)-3-phenylallylidene)amino]-3,3a,4,6a-tetrahydroimidazo[4,5-*d*]imidazol-2(1*H*)-one (5b).** Yield 199 mg (52%); white solid; mp: 134–136 °C. IR (KBr): ν 3081, 3031 (Ar, =CH_2_), 2970, 2932 (Alk), 2877 (C-S), 1978, 1825 (Ar), 1705, 1565 (C=O, C=N) cm^–1^. ^1^H NMR (300 MHz, DMSO-*d*_6_): δ 1.01 (t, *J* = 7.0 Hz, 3H, Me), 1.14 (t, *J* = 7.1 Hz, 3H, Me), 3.10–3.25 (m, 2H, NCH_2_), 3.35–3.55 (m, 2H, NCH_2_), 3.60–3.80 (m, 2H, SCH_2_), 5.12 (d, *J* = 9.9 Hz, 1H, =CH_2_), 5.29 (d, *J* = 17.0 Hz, 1H, =CH_2_), 5.67 (d, *J* = 7.8 Hz, 1H, CH), 5.80–6.10 (m, 2H, CH, -CH=), 6.80–7.10 (m, 2H, =CH, Ph-CH=), 7.20–7.45 (m, 3H, Ph-3–5), 7.60 (d, *J* = 7.5 Hz, 2H, Ph-2,6), 7.79 (d, *J* = 8.0 Hz, 1H, N=CH). ^13^C NMR (75 MHz, DMSO-*d*_6_): δ 13.37 (Me), 13.86 (Me), 32.58 (SCH_2_), 36.21 (NCH_2_), 37.63 (NCH_2_), 70.50 (CH), 78.58 (CH), 118.17 (=CH_2_), 125.24, 126.91, 128.60, 128.80, 133.48, 135.89, 137.60 (=CH, Ph), 140.64 (HC=N), 157.00 (C=O), 164.72 (C=N). HRMS (ESI): Calculated for C_20_H_25_N_5_OS [M + H]^+^: 384.1853; found: 384.1847.

**5-Allylthio-1,3-dimethyl-4-{[(1*E*,2*E*)-3-(2-nitrophenyl)allylidene]amino}-3,3a,4,6a-tetrahydroimidazo[4,5-*d*]imidazol-2(1*H*)-one (5c).** Yield 288 mg (72%); yellow solid; mp: 181–183 °C. IR (KBr): ν 2942, 2908 (Alk), 1699, 1563 (C=O, C=N) cm^–1^. ^1^H NMR (300 MHz, DMSO-*d*_6_): δ 2.81 (s, 3H, NMe), 2.88 (s, 3H, NMe), 3.60–3.80 (m, 2H, SCH_2_), 5.13 (d, *J* = 10.0 Hz, 1H, =CH_2_), 5.29 (d, *J* = 17.0 Hz, 1H, =CH_2_), 5.58 (d, *J* = 7.7 Hz, 1H, CH), 5.80–6.10 (m, 2H, CH, -CH=), 7.03 (dd, *J* = 15.7, 8.9 Hz, 1H, =CH), 7.33 (d, *J* = 15.8 Hz, 1H, Ph-CH=), 7.55 (t, *J* = 7.7 Hz, 1H, Ar), 7.70 (t, *J* = 7.6 Hz, 1H, Ar), 7.96 (d, *J* = 8.0 Hz, 1H, Ar), 8.01–8.10 (m, 2H, Ar, N=CH). ^13^C NMR (75 MHz, DMSO-*d*_6_): δ 28.22 (NMe), 30.45 (NMe), 32.58 (SCH_2_), 71.80 (CH), 79.82 (CH), 118.28 (=CH_2_), 124.34, 127.93, 129.25, 130.15, 130.27, 130.59, 133.28, 133.35 (=CH, Ar), 140.23 (HC=N), 147.91 (ArC-NO_2_), 157.53 (C=O), 164.88 (C=N). HRMS (ESI): Calculated for C_18_H_20_N_6_O_3_S [M + H]^+^: 401.1390; found: 401.1388.

**5-Allylthio-1,3-diethyl-4-{[(1*E*,2*E*)-3-(2-nitrophenyl)allylidene]amino}-3,3a,4,6a-tetrahydroimidazo[4,5-*d*]imidazol-2(1*H*)-one (5d).** Yield 300 mg (70%); yellow solid; mp: 160–162 °C. IR (KBr): ν 3070, 3022 (Ar, =CH_2_), 2970, 2931, 2872 (Alk), 1959, 1881, 1819 (Ar), 1696, 1563 (C=O, C=N) cm^–1^. ^1^H NMR (300 MHz, DMSO-*d*_6_): δ 0.98 (t, *J* = 6.6 Hz, 3H, Me), 1.14 (t, *J* = 6.9 Hz, 3H, Me), 3.05–3.30 (m, 2H, CH_2_), 3.35–3.55 (m, 2H, CH_2_), 3.60–3.80 (m, 2H, SCH_2_), 5.12 (d, *J* = 9.9 Hz, 1H, =CH_2_), 5.29 (d, *J* = 17.0 Hz, 1H, =CH_2_), 5. 68 (d, *J* = 7.6 Hz, 1H, CH), 5.85–6.10 (m, 2H, CH, -CH=), 7.05 (dd, *J* = 15.7, 8.8 Hz, 1H, =CH), 7.31 (d, *J* = 15.8 Hz, 1H, Ph-CH=), 7.55 (t, *J* = 7.6 Hz, 1H, Ar), 7.70 (t, *J* = 7.5 Hz, 1H, Ar), 7.87 (d, *J* = 8.8 Hz, 1H, N=CH), 7.96 (d, *J* = 8.0 Hz, 1H, Ar), 8.03 (d, *J* = 7.8 Hz, 1H, Ar). ^13^C NMR (75 MHz, DMSO-*d*_6_): δ 13.38 (Me), 13.93 (Me), 32.59 (SCH_2_), 36.21 (NCH_2_), 37.73 (NCH_2_), 70.48 (CH), 78.58 (CH), 118.24, 124.39, 128.01, 129.27, 130.09, 130.21, 130.74, 133.33, 133.42 (=CH, Ar), 139.79 (HC=N), 147.86 (ArC-NO_2_), 157.01 (C=O), 164.45 (C=N). HRMS (ESI): Calculated for C_20_H_24_N_6_O_3_S [M + H]^+^: 429.1703; found: 429.1695.

**5-Allylthio-4-{[(1*E*,2*E*)-3-(2-methoxyphenyl)allylidene]amino}-1,3-dimethyl-3,3a,4,6a-tetrahydroimidazo[4,5-*d*]imidazol-2(1*H*)-one (5e).** Yield 300 mg (78%); white solid; mp: 154–156 °C. IR (KBr): ν 3060 (Ar, =CH_2_), 2988, 2969, 2929, 2889, 2837 (Alk, C-S), 1706, 1562 (C=O, C=N) cm^–1^. ^1^H NMR (300 MHz, DMSO-*d*_6_): δ 2.81 (s, 3H, NMe), 2.88 (s, 3H, NMe), 3.60–3.75 (m, 2H, SCH_2_), 3.85 (s, 3H, OMe), 5.12 (d, *J* = 10.0 Hz, 1H, =CH_2_), 5.29 (d, *J* = 16.9 Hz, 1H, =CH_2_), 5.57 (d, *J* = 7.7 Hz, 1H, CH), 5.80–6.10 (m, 2H, CH, -CH=), 6.80–6.99 (m, 2H, Ar, =CH), 7.05 (d, *J* = 8.4 Hz, 1H, Ar), 7.20–7.40 (m, 2H, Ar, Ph-CH=), 7.64 (d, *J* = 7.5 Hz, 1H, Ar), 7.99 (d, *J* = 9.0 Hz, 1H, N=CH). ^13^C NMR (75 MHz, DMSO-*d*_6_): δ 28.23 (NMe), 30.45 (NMe), 32.54 (SCH_2_), 55.48 (OMe), 71.89 (CH), 79.76 (CH), 111.50, 118.20, 120.71, 124.28, 125.68, 126.93, 129.98, 132.34, 133.44 (=CH, Ar), 141.99 (HC=N), 156.64 (ArC-OMe), 157.58 (C=O), 165.09 (C=N). HRMS (ESI): Calculated for C_19_H_23_N_5_O_2_S [M + H]^+^: 386.1645; found: 386.1640.

**5-Allylthio-1,3-diethyl-4-{[(1*E*,2*E*)-3-(2-methoxyphenyl)allylidene]amino}-3,3a,4,6a-tetrahydroimidazo[4,5-*d*]imidazol-2(1*H*)-one (5f).** Yield 330 mg (80%); white solid; mp: 132–135 °C. IR (KBr): ν 3075, 3047 (Ar, =CH_2_), 2969, 2932 (Alk), 2837 (C-S), 1703, 1575 (C=O, C=N) cm^–1^. ^1^H NMR (300 MHz, DMSO-*d*_6_): δ 0.99 (t, *J* = 7.0 Hz, 3H, Me), 1.14 (t, *J* = 7.1 Hz, 3H, Me), 3.08–3.30 (m, 3H, NCH_2_), 3.35–3.50 (m, 1H, NCH_2_), 3.60–3.79 (m, 2H, SCH_2_), 3.85 (s, 3H, OMe), 5.12 (d, *J* = 10.1 Hz, 1H, =CH_2_), 5.29 (d, *J* = 16.9 Hz, 1H, =CH_2_), 5.67 (d, *J* = 7.8 Hz, 1H, CH), 5.80–6.05 (m, 2H, CH, -CH=), 6.90–6.99 (m, 2H, Ar, =CH), 7.05 (d, *J* = 8.3 Hz, 1H, Ar), 7.20–7.40 (m, 2H, Ar, Ph-CH=), 7.65 (d, *J* = 7.4 Hz, 1H, Ar), 7.85 (d, *J* = 9.0 Hz, 1H, N=CH). ^13^C NMR (75 MHz, DMSO-*d*_6_): δ 13.35 (Me), 13.85 (Me), 32.54 (SCH_2_), 36.16 (NCH_2_), 37.58 (NCH_2_), 55.47 (OMe), 70.43 (CH), 78.49 (CH), 111.48, 118.14, 120.68, 124.21, 125.64, 127.02, 129.97, 132.23, 133.48 (=CH, Ar), 141.48 (HC=N), 156.63 (ArC-OMe), 156.99 (C=O), 164.65 (C=N). HRMS (ESI): Calculated for C_21_H_27_N_5_O_2_S [M + H]^+^: 414.1958; found: 414.1951.

**5-Butylthio-1,3-dimethyl-4-[((1*E*,2*E*)-3-phenylallylidene)amino]-3,3a,4,6a-tetrahydroimidazo[4,5-*d*]imidazol-2(1*H*)-one (6a).** Yield 356 mg (96%); white solid; mp: 101–103 °C. IR (KBr): ν 3081, 3057, 3029 (Ar), 2997, 2956, 2928 (Alk), 2871 (C-S), 2053, 1952, 1883, 1823 (Ar), 1701, 1562 (C=O, C=N) cm^–1^. ^1^H NMR (300 MHz, DMSO-*d*_6_): δ 0.91 (t, *J* = 7.3 Hz, 3H, Me), 1.30–1.50 (m, 2H, CH_2_), 1.55–1.80 (m, 2H, CH_2_), 2.83 (s, 3H, NMe), 2.91 (s, 3H, NMe), 3.03 (t, *J* = 7.1 Hz, 2H, SCH_2_), 5.57 (d, *J* = 7.7 Hz, 1H, CH), 5.89 (d, *J* = 7.9 Hz, 1H, CH), 6.95 (dd, *J* = 16.0, 8.5 Hz, 1H, =CH), 7.07 (d, *J* = 16.0 Hz, 1H, Ph-CH=), 7.25–7.50 (m, 3H, Ph-3-5), 7.60 (d, *J* = 7.4 Hz, 2H, Ph-2,6), 7.95 (d, *J* = 8.6 Hz, 1H, N=CH). ^13^C NMR (75 MHz, DMSO-*d*_6_): δ 13.45 (Me), 21.34 (CH_2_), 28.26 (CH_2_), 29.57 (SCH_2_), 30.61 (NMe), 30.76 (NMe), 71.93 (CH), 79.76 (CH), 125.40, 126.87, 127.14, 128.57, 128.82, 135.96, 137.58 (=CH, Ph), 141.01 (HC=N), 157.62 (C=O), 165.65 (C=N). HRMS (ESI): Calculated for C_19_H_25_N_5_OS [M + H]^+^: 372.1853; found: 372.1852.

**5-Butylthio-1,3-diethyl-4-[((1*E*,2*E*)-3-phenylallylidene)amino]-3,3a,4,6a-tetrahydroimidazo[4,5-*d*]imidazol-2(1*H*)-one (6b).** Yield 351 mg (88%); white solid; mp: 103–105 °C. IR (KBr): ν 3079, 3056, 3039 (Ar), 2963, 2928 (Alk), 2870 (C-S), 1938, 1873 (Ar), 1699, 1560 (C=O, C=N) cm^–1^. ^1^H NMR (300 MHz, DMSO-*d*_6_): δ 0.89 (t, *J* = 7.3 Hz, 3H, Me), 1.02 (t, *J* = 7.1 Hz, 3H, Me), 1.14 (t, *J* = 7.1 Hz, 3H, Me), 1.30–1.50 (m, 2H, CH_2_), 1.55–1.75 (m, 2H, CH_2_), 2.95–3.10 (m, 2H, SCH_2_), 3.15–3.30 (m, 3H, NCH_2_), 3.40–3.60 (m, 1H, NCH_2_), 5.66 (d, *J* = 7.8 Hz, 1H, CH), 5.94 (d, *J* = 8.0 Hz, 1H, CH), 6.90–7.10 (m, 2H, =CH, Ph-CH=), 7.30–7.45 (m, 3H, Ph-3-5), 7.61 (d, *J* = 7.1 Hz, 2H, Ph-2,6), 7.79 (d, *J* = 7.8 Hz, 1H, N=CH). ^13^C NMR (75 MHz, DMSO-*d*_6_): δ 13.46 (2Me), 13.90 (Me), 21.38 (CH_2_), 29.59 (CH_2_), 30.88 (SCH_2_), 36.29 (NCH_2_), 37.66 (NCH_2_), 70.39 (CH), 78.65 (CH), 125.33, 126.92, 128.61, 128.83, 135.94, 137.48 (=CH, Ph), 140.48 (HC=N), 157.05 (C=O), 165.24 (C=N). HRMS (ESI): Calculated for C_21_H_29_N_5_OS [M + H]^+^: 400.2166; found: 400.2164.

**5-Butylthio-1,3-dimethyl-4-{[((1*E*,2*E*)-3-(2-nitrophenyl)allylidene]amino}-3,3a,4,6a-tetrahydroimidazo[4,5-*d*]imidazol-2(1*H*)-one (6c).** Yield 403 mg (97%); yellow solid; mp: 148–150 °C. IR (KBr): ν 3081, 3014 (Ar), 2960, 2933 (Alk), 2872 (C-S), 1840, 1766 (Ar), 1696, 1560 (C=O, C=N) cm^–1^. ^1^H NMR (300 MHz, DMSO-*d*_6_): δ 0.90 (t, *J* = 7.3 Hz, 3H, Me), 1.30–1.50 (m, 2H, CH_2_), 1.55–1.75 (m, 2H, CH_2_), 2.82 (s, 3H, NMe), 2.89 (s, 3H, NMe), 3.03 (t, *J* = 7.2 Hz, 2H, SCH_2_), 5.57 (d, *J* = 7.7 Hz, 1H, CH), 5.91 (d, *J* = 8.0 Hz, 1H, CH), 7.04 (dd, *J* = 15.8, 8.8 Hz, 1H, =CH), 7.33 (d, *J* = 15.8 Hz, 1H, Ph-CH=),7.55 (t, *J* = 7.2 Hz, 1H, Ar), 7.71 (t, *J* = 7.1 Hz, 1H, Ar), 7.96 (d, *J* = 8.1 Hz, 1H, Ar), 8.01–8.10 (m, 2H, Ar, N=CH). ^13^C NMR (75 MHz, DMSO-*d*_6_): δ 13.42 (Me), 21.30 (CH_2_), 28.23 (CH_2_), 29.59 (SCH_2_), 30.48 (NMe), 30.70 (NMe), 71.67 (CH), 79.75 (CH), 124.33, 127.90, 129.22, 130.16, 130.32, 130.44. 133.28, 135.82 (=CH, Ar), 140.05 (HC=N), 147.88 (ArC-NO_2_), 157.56 (C=O), 165.40 (C=N). HRMS (ESI): Calculated for C_19_H_24_N_6_O_3_S [M + H]^+^: 417.1703; found: 417.1701.

**5-Butylthio-1,3-diethyl-4-{[(1*E*,2*E*)-3-(2-nitrophenyl)allylidene]amino}-3,3a,4,6a-tetrahydroimidazo[4,5-*d*]imidazol-2(1*H*)-one (6d).** Yield 417 mg (94%); yellow solid; mp: 135–137 °C. IR (KBr): ν 3059 (Ar), 2957, 2930 (Alk), 2870 (C-S), 1692, 1566 (C=O, C=N) cm^–1^. ^1^H NMR (300 MHz, DMSO-*d*_6_): δ 0.89 (t, *J* = 7.3 Hz, 3H, Me), 0.99 (t, *J* = 7.0 Hz, 3H, Me), 1.15 (*J* = 7.1 Hz, 3H, Me), 1.30–1.50 (m, 2H, CH_2_), 1.55–1.75 (m, 2H, CH_2_), 2.90–3.10 (m, 2H, SCH_2_), 3.25–3.30 (m, 3H, NCH_2_), 3.40–3.55 (m, 1H, NCH_2_), 5.67 (d, *J* = 7.8 Hz, 1H, CH), 5.96 (d, *J* = 8.1 Hz, 1H, CH), 7.06 (dd, *J* = 15.8, 8.8 Hz, 1H, =CH), 7.31 (d, *J* = 15.8 Hz, 1H, Ph-CH=), 7.56 (t, *J* = 7.1 Hz, 1H, Ar), 7.72 (t, *J* = 7.2 Hz, 1H, Ar), 7.87 (d, *J* = 8.9 Hz, 1H, N=CH), 7.98 (d, *J* = 8.1 Hz, 1H, Ar), 8.03 (d, *J* = 7.1 Hz, 1H, Ar). ^13^C NMR (75 MHz, DMSO-*d*_6_): δ 13.44 (2Me), 13.93 (Me), 21.35 (Me), 29.60 (CH_2_), 30.83 (CH_2_), 36.27 (NCH_2_), 37.76 (NCH_2_), 70.36 (CH), 78.63 (CH), 124.40, 127.99, 129.26, 130.16, 130.24, 130.59, 133.34 (=CH, Ar), 139.61 (HC=N), 147.86 (ArC-NO_2_), 157.04 (C=O), 164.95 (C=N). HRMS (ESI): Calculated for C_21_H_28_N_6_O_3_S [M + H]^+^: 445.2016; found: 445.2013.

**5-Butylthio-4-{[(1*E*,2*E*)-3-(2-methoxyphenyl)allylidene]amino}-1,3-dimethyl-3,3a,4,6a-tetrahydroimidazo[4,5-*d*]imidazol-2(1*H*)-one (6e).** Yield 349 mg (87%); white solid; mp: 113–115 °C. IR (KBr): ν 3074, 3045 (Ar), 2998, 2957, 2928 (Alk), 2871 (C-S), 1711, 1562 (C=O, C=N) cm^–1^. ^1^H NMR (300 MHz, DMSO-*d*_6_): δ 0.91 (t, *J* = 7.3 Hz, 3H, Me), 1.30–1.50 (m, 2H, CH_2_), 1.55–1.75 (m, 2H, CH_2_), 2.82 (s, 3H, NMe), 2.90 (s, 3H, NMe), 3.03 (t, *J* = 7.1 Hz, 2H, SCH_2_), 3.87 (s, 3H, OMe), 5.57 (d, *J* = 7.8 Hz, 1H, CH), 5.90 (d, *J* = 7.9 Hz, 1H, CH), 6.85–6.99 (m, 2H, Ar, =CH), 7.07 (d, *J* = 8.3 Hz, 1H, Ar), 7.20–7.40 (m, 2H, Ar, Ph-CH=), 7.67 (d, *J* = 7.2 Hz, 1H, Ar), 8.01 (d, *J* = 9.0 Hz, 1H, N=CH). ^13^C NMR (75 MHz, DMSO-*d*_6_): δ 13.48 (Me), 21.36 (CH_2_), 28.27 (CH_2_), 29.57 (SCH_2_), 30.50 (NMe), 30.77 (NMe), 55.48 (OMe), 71.76 (CH), 79.71 (CH), 111.49, 120.73, 124.30, 125.76, 126.93, 129.98, 132.21 (=CH, Ar), 141.83 (HC=N), 156.63 (ArC-OMe), 157.62 (C=O), 165.62 (C=N). HRMS (ESI): Calculated for C_20_H_27_N_5_O_2_S [M + H]^+^: 402.1958; found: 402.1954.

**5-Butylthio-1,3-diethyl-4-{[(1*E*,2*E*)-3-(2-methoxyphenyl)allylidene]amino}-3,3a,4,6a-tetrahydroimidazo[4,5-*d*]imidazol-2(1*H*)-one (6f**). Yield 408 mg (95%); white solid; mp: 108–110 °C. IR (KBr): ν 3072, 3044 (Ar), 2960, 2932 (Alk), 2872 (C-S), 2022, 1899, 1824 (Ar), 1699, 1562 (C=O, C=N) cm^–1^. ^1^H NMR (300 MHz, DMSO-*d*_6_): δ 0.90 (t, *J* = 7.3 Hz, 3H, Me), 1.01 (t, *J* = 7.0 Hz, 3H, Me), 1.15 (t, *J* = 7.1 Hz, 3H, Me), 1.30–1.50 (m, 2H, CH_2_), 1.55–1.75 (m, 2H, CH_2_), 2.90–3.10 (m, 2H, SCH_2_), 3.15–3.30 (m, 2H, NCH_2_), 3.40–3.55 (m, 2H, NCH_2_), 3.87 (s, 3H, OMe), 5.67 (d, *J* = 7.8 Hz, 1H, CH), 5.95 (d, *J* = 7.9 Hz, 1H, CH), 6.90–7.01 (m, 2H, Ar, =CH), 7.07 (d, *J* = 8.3 Hz, 1H, Ar), 7.20–7.40 (m, 2H, Ar, Ph-CH=), 7.67 (d, *J* =7.3 Hz, 1H, Ar), 7.85 (d, *J* = 8.9 Hz, 1H, N=CH). ^13^C NMR (75 MHz, DMSO-*d*_6_): δ 13.44 (2Me), 13.90 (Me), 21.37 (Me), 29.58, 30.87 (SCH_2_), 36.27 (NCH_2_), 37.64 (NCH_2_), 55.51 (OMe), 70.34 (CH), 78.59 (CH), 111.51, 120.73, 124.27, 125.75, 127.06, 129.99, 132.16 (=CH, Ar), 141.33 (HC=N), 156.67 (ArC-OMe), 157.08 (C=O), 165.22 (C=N). HRMS (ESI): Calculated for C_22_H_31_N_5_O_2_S [M + H]^+^: 430.2271; found: 430.2279.

**5-Butylthio-4-{[(1*E*,2*E*)-3-(furan-2-yl)allylidene]amino}-1,3-dimethyl-3,3a,4,6a-tetrahydroimidazo[4,5-*d*]imidazol-2(1*H*)-one (6g).** Yield 321 mg (89%); light brown solid; mp: 130–132 °C. IR (KBr): ν 3116 (Furyl), 2957, 2929 (Alk), 2871 (C-S), 1700, 1569 (C=O, C=N) cm^–1^. ^1^H NMR (300 MHz, DMSO-*d*_6_): δ 0.89 (t, *J* = 7.3 Hz, 3H, Me), 1.30–1.50 (m, 2H, CH_2_), 1.55–1.75 (m, 2H, CH_2_), 2.81 (s, 3H, NMe), 2.88 (s, 3H, NMe), 3.01 (t, *J* = 7.0 Hz, 2H, SCH_2_), 5.55 (d, *J* = 7.7 Hz, 1H, CH), 2.84 (d, *J* = 7.9 Hz, 1H, CH), 6.50–6.80 (m, 3H, Fu, =CH), 6.94 (d, *J* = 15.8 Hz, 1H, Ph-CH=), 7.74 (d, *J* = 1.3 Hz, 1H, Fu), 7.89 (d, *J* = 9.2 Hz, 1H, N=CH). ^13^C NMR (75 MHz, DMSO-*d*_6_): δ 13.44 (Me), 21.32 (CH_2_), 28.25 (CH_2_), 29.54 (SCH_2_), 30.57 (NMe), 30.75 (NMe), 71.96 (CH), 79.75 (CH), 111.01, 112.37, 122.73, 123.20, 124.70, 140.38, 144.03 (=CH, Fu), 151.85 (HC=N), 157.62 (C=O), 165.59 (C=N). HRMS (ESI): Calculated for C_17_H_23_N_5_O_2_S [M + H]^+^: 362.1645; found: 362.1647.

**5-Butylthio-1,3-diethyl-4-{[(1*E*,2*E*)-3-(furan-2-yl)allylidene]amino})-3,3a,4,6a-tetrahydroimidazo[4,5-*d*]imidazol-2(1*H*)-one (6h).** Yield 335 mg (86%); light brown solid; mp: 127–129 °C. IR (KBr): ν 3117 (Furyl), 2960, 2931 (Alk), 2872 (C-S), 1698, 1571 (C=O, C=N) cm^–1^. ^1^H NMR (300 MHz, DMSO-*d*_6_): δ 0.90 (t, *J* = 7.3 Hz, 3H, Me), 1.00 (t, *J* = 6.9 Hz, 3H, Me), 1.15 (t, *J* = 7.0 Hz, 3H, Me), 1.30–1.50 (m, 2H, CH_2_), 1.55–1.75 (m, 2H, CH_2_), 2.90–3.10 (m, 2H, SCH_2_), 3.12–3.30 (m, 3H, NCH_2_), 3.40–3.60 (m, 1H, NCH_2_), 5.67 (d, *J* = 7.8 Hz, 1H, CH), 5.91 (d, *J* = 7.9 Hz, 1H, CH), 6.50–6.58 (m, 3H, Fu, =CH), 6.94 (d, *J* = 15.9 Hz, 1H, Ph-CH=), 7.70–7.80 (m, 2H, Fu, N=CH). ^13^C NMR (75 MHz, DMSO-*d*_6_): δ 13.45 (2Me), 13.87 (Me), 21.36 (Me), 29.54, 30.87, 36.27 (NCH_2_), 37.64 (NCH_2_), 70.42 (CH), 78.63 (CH), 111.12, 112.39, 123.10, 124.62, 139.82, 144.07 (=CH, Fu), 151.83 (HC=N), 157.02 (C=O), 165.14 (C=N). HRMS (ESI): Calculated for C_19_H_27_N_5_O_2_S [M + H]^+^: 390.1958; found: 390.1945.

### 3.2. Bioassays of Fungicidal Activities against Phytopathogenic Fungi

The fungicidal activities were tested according to the conventional procedure [20,23,24,25,26] with six phytopathogenic fungi from different taxonomic classes: *Venturia inaequalis* (*V. i.*), *Rhizoctonia solani* (*R. s.*), *Fusarium oxysporum* (*F. o.*), *Fusarium moniliforme* (*F. m.*), *Bipolaris sorokiniana* (*B. s.*), and *Ssclerotinia sclerotiorum* (*S. s.*). The effect of the chemicals on mycelial radial growth was determined by dissolving a concentration of 3 mg mL^−1^ in acetone and suspending aliquots in potato-saccharose agar at 50 °C to give a concentration of 30 µg mL^−1^. The final acetone concentration of both fungicide-containing and control samples was 10 mL L^−1^. Petri dishes containing 15 mL of the agar medium were inoculated by placing 2 mm mycelial agar discs on the agar surface. Plates were incubated at 25 °C and radial growth was measured after 72 h. The mixed medium without a sample was used as the blank control. Three replicates of each test were carried out. The mycelium elongation diameter (mm) of fungi settlements was measured after 72 h of culture. The growth inhibition rates were calculated with the following equation: I = [(DC − DT)/DC] × 100%. Here, I is the growth inhibition rates (%), DC is the control settlement diameter (mm), and DT is the treatment group fungi settlement diameter (mm). The results are summarized in Table 1.

### 3.3. Antifungal Assay against C. albicans and C. neoformans

Preliminary antifungal screening and MIC determination assays were carried out at the University of Queensland following reported methods [29,30,31]. Fungal strains were cultured in yeast nitrogen base media (YNB), and added to each well of the compound-containing plates (384-well plates, NBS). Plates were covered and incubated at 35 °C for 36 h. Growth inhibition of C. albicans was determined by measuring absorbance at 530 nm (OD_530_), while the growth inhibition of C. neoformans was determined by measuring the difference in absorbance between 600 and 570 nm (OD_600–570_), after the addition of resazurin (0.001% final concentration) and incubation at 35 °C for an additional 2 h. The absorbance was measured using a Biotek Synergy HTX plate reader. The percentage of growth inhibition was calculated for each well, using the negative control (media only) and positive control (bacteria without inhibitors) on the same plate as references. The minimum inhibitory concentration (MIC) was determined following the CLSI guidelines, identifying the lowest concentration at which the full inhibition of the fungi was detected. Full inhibition of growth was defined at <=20% growth (or >80% inhibition), and concentrations were only selected if the next highest concentration displayed full inhibition (i.e., 80–100%) as well (eliminating ‘singlet’ active concentration). In addition, the maximal percentage of growth inhibition was reported as D_Max_, indicating any compounds with marginal activity. Compounds were plated as a 2-fold dose response from 32 to 0.25 μg/mL, with a maximum of 0.5% DMSO in the final assay concentration. Samples with MIC <= 16 μg/mL were declared as a hit.

### 3.4. Cytotoxicity Assay by COADD

Cytotoxicity assays were carried out at the University of Queensland following reported methods [29,30,31]. HEK293 ATCC CRL-1573 human embryonic kidney cells were cultured in Dulbecco’s Modified Eagle Medium (DMEM) with 10% fetal bovine serum (FBS). The cells were incubated together with the compounds for 20 h at 37 °C in 5% CO_2_. Growth inhibition of HEK293 cells was determined by measuring fluorescence at ex: 530/10 nm and em: 590/10 nm (F_560/590_), after the addition of resazurin (25 μg/mL final concentration) and incubation at 37 °C for an additional 3 h in 5% CO_2_. The fluorescence was measured using a Tecan M1000 Pro monochromator plate reader. The percentage of growth inhibition was calculated for each well, using the negative control (media only) and positive control (cell culture without inhibitors) on the same plate as references.

CC_50_ (concentration at 50% cytotoxicity) was calculated by curve fitting the inhibition values vs. log (concentration) using the sigmoidal dose–response function, with variable values for bottom, top, and slope. The curve fitting was implemented using Pipeline Pilot’s dose–response component (giving similar results to similar tools such as GraphPad’s Prism and IDBS’s XlFit). Any value with >indicates a sample with no activity (low D_Max_ value) or samples with CC_50_ values above the maximum tested concentration (higher D_Max_ value).

Cytotoxic samples were classified using CC_50_ ≤ 32 µg mL^−1^ in either replicate (n = 2 in different plates). 

### 3.5. Hemolysis Assay

Hemolytic activity assays were carried out at the University of Queensland using human red blood cells (RBC) following reported methods [29,30,31]. HC_10_ (Concentration at 10% haemolytic activity) and HC_50_ (concentration at 50% haemolytic activity) were calculated by curve fitting the inhibition values versus log (concentration) using a sigmoidal dose–response function with variable fitting values for top, bottom, and slope. The curve fitting was implemented using Pipeline Pilot’s dose–response component (giving similar results to similar tools such as GraphPad’s Prism and IDBS’s XlFit). 

### 3.6. Cytotoxicity assay against 60 Cancer Cell Lines at the National Cancer Institute

The initial assessment of cytotoxic activity was performed using an NCI 60 set of sixty human tumor cell lines derived from nine tumor diseases according to the National Cancer Institute’s Division of Drug Evaluation protocol at a single dose (10^–5^ mol L^–1^) [32]. The percentage growth number is an increase in the number of cells compared to the control, wherein the cells were not treated with the test substance. This allows the detection of both growth inhibition (values from 0 to 100) and lethality (values less than 0). A value of 100 means no growth inhibition.

## 4. Conclusions

The fungicidal activity of a series of S-alkyl substituted N-(3-arylallylidene)aminothioglycolurils was studied, and structure–activity relationships were established. Compounds exceeding or comparable to the well-known fungicidal agent triadimefon were identified. The S-alkyl-N-(3-arylallylidene)aminothioglycolurils with an unsubstituted phenyl or furyl ring were the most effective at inhibiting the growth of phytopathogenic fungi. Inhibitory activity increased with an increase in the length of the alkyl chain at the S atom. Compounds containing an ethyl substituent at the S atom demonstrated elevated activity against yeast *Candida albicans*. The hemo- and cytotoxicity tests showed that 5-ethylthio-1,3-dimethyl-6-[(*E*)-((*E*)-3-phenylallylidene)amino]-3,3a,6,6a-tetrahydroimidazo[4,5-*d*]imidazol-2(1*H*)-one **3a** and 5-ethylthio-6-{(*E*)-[(*E*)-3-(furan-2- yl)allylidene]amino}-1,3-dimethyl-3,3a,6,6a-tetrahydroimidazo[4,5-*d*]imidazol-2(1*H*)-one **3g** exhibited low toxicity to the normal human cell lines, while still showing high fungicidal activity. Further research should focus on *in vivo* assessment of the fungicidal activity of S-alkyl thioglycolurils and clarification of the mechanism of their action.

## Data Availability

The data presented in this study are available in the article and in the Appendix A.

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
