# Peer review of "Synthesis and Evaluation on the Fungicidal Activity of S-Alkyl Substituted Thioglycolurils"

_ijms, 2023, doi:10.3390/ijms24065756_

Round 1
Reviewer 1 Report
This paper deals with the synthesis and evaluation of new fungicide. The paper contains interesting results, however, there are some problems as follows. So, the paper can be published after the following problems are corrected.
— In title, the two words “of” would be caused the prevention of easy understandings. So it would be “… Evaluation on the Fungicidal Activity ….”
— The authors should describe why the authors synthesized the new fungicides. If there was any problems on the previous materials, the authors should show. For the new fungicides synthesis, molecular design concept would be shown, if any.
— In Fig. 3, the data is same to Table 1. Fig. 3 would be omitted.
— The authors should discuss why phenyl or furyl ring are effective.
— The authors would show the perspectives of their research concisely.
Author Response
Thank you very much for your valuable comments. Below are the reviewer's comments and our responses to them. We tried to take into account in the revised version all the comments and suggestions of the reviewers.
Best regards, Galina Gazieva
- In title, the two words “of” would be caused the prevention of easy understandings.So it would be “… Evaluation on the Fungicidal Activity ….”
Corrected.
- The authors should describe why the authors synthesized the new fungicides.If there was any problems on the previous materials, the authors should show.For the new fungicides synthesis, molecular design concept would be shown, if any.
We added in the manuscript some discussion: Recently [20] we have synthesized a series of S-methyl and S-ethyl thioglycoluril derivatives 2, 3 possessing fungicidal activity against phytopatogenes Rhizoctonia solani, Fusarium oxysporum, Fusarium moniliforme and Bipolaris sorokiniana (see Figure 1). It was found that S-ethyl derivatives are somewhat more active than S-methyl ones. We assumed that the elongation of the alkyl chain at the S atom to C3 and C4 would lead to an increase in fungicidal activity. In this paper, we synthesized a series of new S-propyl, S-allyl and S-butyl derivatives of 4-[(E)-((E)-3-phenylallylidene)amino]-5-thioxohexahydroimidazo[4,5- d]imidazol-2(1H)-one (4-[(E)-((E)-3-phenylallylidene)amino]thioglycoluril) and evaluated their fungicidal activity. S-Allyl thioglycolurils were synthesized to reveal the influence of unsaturated fragment on the activity.
We also added the aimed compounds at Fig. 1.
- In Fig. 3, the data is same to Table 1.Fig. 3 would be omitted.
We would like to keep Figure 3 for clarity.
- The authors should discuss why phenyl or furyl ring are effective.
We added in the manuscript the sentence: Perhaps, besides electron effects of the aryl ring substituents, the steric ones can affect the activity.
- The authors would show the perspectives of their research concisely.
We added in the Conclusion the sentence: Further research should focus on in vivo assessing the fungicidal activity of S-alkyl thioglycolurils.
Reviewer 2 Report
Vinogradova et al. reported the synthesis and evaluation of the fungicidal activity of S-alkyl 2 substituted thioglycolurils.
I have some minor concerns that need to be addressed before publication.
1. Designing of thioglycolurils as antifungals should be discussed in detail.
2. SAR should be revised; a conclusive SAR should be presented.
3. The mechanism of action of these compounds should also be discussed.
4. Typo errors should be addressed.
5. The conclusion part should also be revised.
Author Response
Thank you very much for your valuable comments. Below are the reviewer's comments and our responses to them. We tried to take into account in the revised version all the comments and suggestions of the reviewers.
Best regards, Galina Gazieva
- Designing of thioglycolurils as antifungals should be discussed in detail.
We added in the manuscript some discussion: Recently [20] we have synthesized a series of S-methyl and S-ethyl thioglycoluril derivatives 2, 3 possessing fungicidal activity against phytopatogenes Rhizoctonia solani, Fusarium oxysporum, Fusarium moniliforme and Bipolaris sorokiniana (see Figure 1). It was found that S-ethyl derivatives are somewhat more active than S-methyl ones. We assumed that the elongation of the alkyl chain at the S atom to C3 and C4 would lead to an increase in fungicidal activity. In this paper, we synthesized a series of new S-propyl, S-allyl and S-butyl derivatives of 4-[(E)-((E)-3-phenylallylidene)amino]-5-thioxohexahydroimidazo[4,5- d]imidazol-2(1H)-one (4-[(E)-((E)-3-phenylallylidene)amino]thioglycoluril) and evaluated their fungicidal activity. S-Allyl thioglycolurils were synthesized to reveal the influence of unsaturated fragment on the activity.
We also added the aimed compounds at Fig. 1.
- SAR should be revised; a conclusive SAR should be presented.
We added in the manuscript some structure-activity correlations: (i) an increase of the length of the alkyl chain at the sulfur atom leads to an increase of activity; (ii) depending on arylmethylidene fragments, compounds could be arranged according to decreasing activity as follows: Ph-substituted ≥ Fu-substituted > 2-MeOC6H4-substituted >> 2-NO2C6H4- substituted.
- The mechanism of action of these compounds should also be discussed.
We assume that the mechanism of action of these compounds is the same as the mechanism of action of fenamidone. However, it has not yet been proved.
- Typo errors should be addressed.
We made some corrections.
- The conclusion part should also be revised.
We added in the Conclusion the sentence: Further research should focus on in vivo assessing the fungicidal activity of S-alkyl thioglycolurils and clarifying the mechanism of their action.
Reviewer 3 Report
In this manuscript, Gazieva and coworkers synthesized a series of S-alkyl substituted thioglycolurils through alkylation of corresponding thioglycolurils and tested their fungicidal activity against multiple phytopathogenic fungus from different taxonomic classes. Several S-alkyl substituted thioglycolurils exhibited high mycelium growth inhibition in bioassays. Later, they revealed that S-ethyl substituted thioglycolurils possessed high inhibition activity against Candida albicans and their cytotoxicity was tested against HEK-293 and RBC cell lines. Considering the importance and novelty of this work, this manuscript should be accepted for publication after addressing the following comments.
1. Mycelium growth inhibition data is missing for compounds 4f, 5a and 5b. This data should be added into table 1, even though negative results.
2. Authors explored cytotoxic profile for compounds 3a, 3b, 3e, 3g. But none of the newly synthesized compounds (4-6) is tested in cytotoxic assay. At least a few potent compounds should be investigated.
3. Line 131-132: “All the tested compounds did not possess activity against Cryptococcus neoformans in concentration 32 μg-mL-1”. Does author have any rationale why these compounds are completely inactive against Cryptococcus neoformans even though showed activity against Candida albicans.
4. Data for Reference Compound is missing for Table 2.
5. In Supplementary, LC-MS data of 4f : What is the wavelength of UV lamp used here? In DAD1 UV trace, multiple small peaks are shown along with large, desired peak at 1.448. Some of those small peaks indicate mass ions of 415, 416, and 417, which corresponds to molecular ion of 4f. What is the reason for showing these isomeric peaks? Are these because of fluxional characteristic of the molecule (sp3 N-atoms) or diastereomeric interconversion along C=N and C=C bond?
6. A particular format for describing fungus taxonomic classes should be followed. It should be italicized in every case.

Author Response
Thank you very much for your valuable comments. Below are the reviewer's comments and our responses to them. We tried to take into account in the revised version all the comments and suggestions of the reviewers.
Best regards, Galina Gazieva
- Mycelium growth inhibition data is missing for compounds 4f, 5a and 5b. This data should be added into table 1, even though negative results.
Unfortunately, we can’t add these data. Compound 4f was not tested because of it was only 95% purity according to LS/MS data. For compounds 5a and 5b, poor reproducible data were obtained.
- Authors explored cytotoxic profile for compounds 3a, 3b, 3e, 3g. But none of the newly synthesized compounds (4-6) is tested in cytotoxic assay. At least a few potent compounds should be investigated.
We added cytotoxicity data for new compounds 4b and 5d-f obtained by the National Cancer Institute Developmental Therapeutic Program (DTP). The compounds practically do not possess cytotoxicity.
- Line 131-132: “All the tested compounds did not possess activity against Cryptococcus neoformans in concentration 32 μg-mL-1”. Does author have any rationale why these compounds are completely inactive against Cryptococcus neoformans even though showed activity against Candida albicans.
This sentence was rephrase and we added some data: All the tested compounds did not possess significant activity against Cryptococcus neoformans in concentration 32 μg mL-1 (< 50% mycelium growth inhibition, MIC values >32 μg mL-1 for 3a,b,e,g).
- Data for Reference Compound is missing for Table 2.
The Community for Antimicrobial Drug Discovery (COADD ) did not supply the data for Reference Compound. Two values are used by COADD as quality controls for individual plates: Z'-Factor=[1 -(3*(sd(NegCtrl)+sd(PosCtrl))/(average(PosCtrl)-average(NegCtrl))) ] and Standard Antibiotic controls at different concentrations (>MIC and < MIC). The plate passes the quality control if Z'-Factor >0.4 and Standards are active and inactive at highest and lowest concentrations, respectively. Data not supplied.
We added in Table 2 data for Fluconazole reported in the literature [31].
- In Supplementary, LC-MS data of 4f : What is the wavelength of UV lamp used here? In DAD1 UV trace, multiple small peaks are shown along with large, desired peak at 1.448. Some of those small peaks indicate mass ions of 415, 416, and 417, which corresponds to molecular ion of 4f. What is the reason for showing these isomeric peaks? Are these because of fluxional characteristic of the molecule (sp3 N-atoms) or diastereomeric interconversion along C=N and C=C bond?
The wavelength of the UV lamp used has been added. Isomeric peaks have been removed.
- A particular format for describing fungus taxonomic classes should be followed. It should be italicized in every case.
Corrected.